# Endoscopy to Diagnose and Prevent Digestive Cancers in Lynch Syndrome

**DOI:** 10.3390/cancers13143505

**Published:** 2021-07-13

**Authors:** Raphael Olivier, Violaine Randrian, David Tougeron, Jean-Christophe Saurin

**Affiliations:** 1Gastroenterology Department, Poitiers University Hospital (CHU de Poitiers), 86000 Poitiers, France; violaine.randrian@chu-poitiers.fr (V.R.); david.tougeron@chu-poitiers.fr (D.T.); 2Gastroenterology Department, Hospices Civils de Lyon—Centre Hospitalier Universitaire, 69002 Lyon, France; jean-christophe.saurin@chu-lyon.fr

**Keywords:** Lynch syndrome, digestive endoscopy, colorectal cancer

## Abstract

**Simple Summary:**

Lynch syndrome is characterized by a higher relative risk of developing certain cancers, especially digestive cancers. Many guidelines from different scientific societies are now available and allow excellent follow-up for these patients, but occasionally propose divergent management approaches. We provide here a synthesis of these guidelines and a focus on prevention, diagnosis, and endoscopic follow-up of these digestive cancers and early neoplasia.

**Abstract:**

Lynch syndrome patients could benefit from various recommendations to prevent digestive cancers. In this review, we summarize the criteria to identify Lynch syndrome in patients with digestive cancers. We detail endoscopic screening procedures in patients with Lynch syndrome for gastric, small bowel, pancreatic, and colorectal cancers. We review the precise modalities of endoscopic follow-up, particularly the discrepancies that exist between the guidelines of the various scientific societies. We discuss the treatment of colorectal cancers in Lynch syndrome cases and patient adherence to endoscopic follow-up programs.

## 1. Introduction

The large majority of colorectal cancers (CRCs) are sporadic. Nevertheless, different studies have indicated that 35% of CRC cases have a familial component [1], and in 2–5% of CRC cases, a genetic origin can be identified [2]. Lynch syndrome (LS) is the most common hereditary colorectal cancer syndrome, with an estimated population frequency of 1/280 individuals [3,4]. LS represents about 3% of all CRCs [5]. It is linked to a germline mutation in one of the DNA mismatch repair (MMR) system genes (*MLH1*, *MSH2*, *MSH6,* or *PMS2* or a deletion in the 3′ region of the epithelial cell adhesion molecule (*EPCAM)* gene 4) [2]. The inactivation of the MMR system leads to errors during the replication of repeated DNA sequences, called microsatellites, resulting in microsatellite instability (MSI).

Determination of MMR deficiency (dMMR) and MSI status has a major impact on CRC management, notably in Lynch syndrome follow-up, and has major prognostic and predictive value [5]. MMR deficiency/MSI status is associated with a better prognosis in patients with non-metastatic CRC; that said, stage II dMMR/MSI CRCs show chemoresistance to adjuvant 5-fluorouracil alone [6,7]. By contrast, these patients are sensitive to adjuvant oxaliplatine-based chemotherapy [8]. At a metastatic stage, however, dMMR/MSI CRCs are associated with very poor survival and chemoresistance [9]. Nevertheless, dMMR/MSIs at both non-metastatic and metastatic stages are very sensitive to immune checkpoint inhibitors [10]. In other digestive tumors, primarily gastric and small bowel adenocarcinoma, MSI status has been associated with good prognosis and high sensitivity to immune checkpoint inhibitors [11]. Finally, in LS families, endoscopic follow-up is a major issue in efforts to remove pre-cancerous lesions, prevent cancer, and allow early detection of neoplasia.

In this review, we propose to clarify the clinical criteria identifying Lynch syndrome in patients with digestive cancers. We will then summarize the recommended endoscopic follow-up in this syndrome. While many guidelines are now available, discrepancies may appear, confusing the clinician. We thus propose a synthesis of the recommendations, with the objective being to determine the most efficient and consensual management of the patient.

## 2. Criteria to Identify and Diagnose Lynch Syndrome

### 2.1. Clinical Suspicion of Lynch Syndrome

It should be recalled that LS is characterized by a higher relative risk of developing specific cancers. The most common ones include CRC, endometrial, small intestine, and urothelial carcinoma (involving the upper urinary tract); rarer cancers are also possible: ovarian cancers, glioblastomas, sebaceous skin tumors, cholangiocarcinomas, and gastric adenocarcinomas. LS should be suspected if a patient presents one or several cancers from these spectrums, especially at a young age. We will focus in this review on digestive cancers, owing to their screening eligibility, especially colorectal cancer.

Some clinical criteria may help to identify LS. The Bethesda criteria revised in 2004 are probably the most useful thanks to their simplicity of use and their precise description (Table 1) [12]. In contrast, the Amsterdam II criteria are clearly too restrictive with a high rate of false negatives [13], despite their specificity. Too many patients were not diagnosed because they did not necessarily meet the four criteria. In clinical practice, colorectal cancer patients with family history meeting either the Amsterdam II criteria and/or the revised Bethesda criteria should receive systematic MSI testing and/or IHC for MMR protein expression and probably be referred for oncogenetic consultation.

The European recommendations encourage tumor screening for MSI and/or loss of expression of MMR proteins in immunohistochemistry (IHC) for any patient under 70 years of age presenting a first CRC [14] without taking into consideration the spectrum of the other cancers. The recommendations vary from country to country; for example, in the American [15,16] and British recommendations [17], the search for MSI and/or loss of MMR protein expression in IHC is recommended for all initially diagnosed CRCs (Table 2). In the case of high suspicion of LS, i.e., using revised Bethesda criteria, CRC patients must have MMR IHC/MSI tests. Nevertheless, because of the major therapeutic impact of dMMR/MSI status, universal screening of all CRCs can be proposed.

### 2.2. Somatic Analyses of Colorectal Lesions in the Case of Suspected Lynch Syndrome

In the case of suspicion of LS, it is first required to confirm MSI and loss of expression of MMR proteins by IHC. These somatic tests are performed on the tumor, and dMMR phenotype is defined by nuclear loss of expression of one or more MMR proteins. As they function as dimers, mutation generally concerns two proteins, even if only one is mutated (*MLH1/PMS2* or *MSH2/MSH6*). This method also suggests the mutated gene corresponding to the lost proteins [18]. MSI phenotype is determined by polymerase chain reaction using a panel of five microsatellite mononucleotide markers, called Pentaplex (BAT-25, BAT-26, NR-21, NR-22, and NR-24 or BAT-25, BAT-26, D5S346, D2S123, and D17S250) [19,20]. Instability of at least three markers defines MSI status. dMMR/MSI CRCs with no *MLH1* promoter hypermethylation and/or *BRAF* mutation should have genetic counseling MMR germline testing to confirm LS. Lynch-like syndrome (LLS) has been proposed for patients with a family history of cancers associated with Lynch syndrome, but no pathogenic germline MMR mutation [21]. According to family history and oncogenetician recommendations, these patients should undergo endoscopic screening.

Both methods could be used, but some studies have shown discordances between MMR protein IHC and MSI molecular testing, with rates ranging from 1% to 10% [22,23,24]. Expert centers usually perform both tests. That said, these tests are expensive, with limited availability, and time- consuming. It is consequently recommended to use only one test as universal screening of all CRC patients with no suspicion of LS [25]. By contrast, two tests should be performed in the event of high suspicion of LS (e.g., multiple LS cancer spectrums in the family) [18,26]. The major impact in the case of proven LS justifies the propensity to perform both tests in order not to avoid missing out on LS.

It is worth noting that, whenever possible, these tests should be performed on cancers and not on adenomas, as false negative results exist [27]. Indeed, the combination of MSI and IHC testing in colorectal adenomas is useful only when LS is suspected and adenomatous polyps are the only tissues available for analysis. A negative result does not exclude the presence of LS, especially in the case of low-grade dysplasia adenomas.

In conclusion, screening for MSI and/or loss of expression of MMR proteins by IHC for all CRCs represents to an ever greater extent a gold standard and should be widely proposed.

## 3. Endoscopic Follow-Up for Patients with Lynch Syndrome

Gastroenterology societies have published guidelines for high-risk CRC patients screening and follow-up, especially the British Society of Gastroenterology (BSG), the Association of Coloproctology of Great Britain and Ireland (ACPGBI), the United Kingdom Cancer Genetics Group (UKCGG) [17], and the European Society of Gastrointestinal Endoscopy (ESGE) Guidelines [28].

### 3.1. Follow-Up in Expert Networks/Centers

One major point is that patients with LS should be followed in expert centers with specialized networks. The efficacy of colonoscopy follow-up and adenoma resection on the incidence and mortality of CRC in LS patients is well-known [29,30,31]. As this monitoring is lifelong and difficult from a psychological point of view, patient adherence is essential. Studies have demonstrated the efficacy and benefit of cancer risk education [32] and standardized surveillance programs [33,34] to improve compliance to colonoscopy in LS patients. Inclusion in national registries, associated with endoscopic surveillance programs and the use of reminders, results in high compliance rates [25,35,36,37].

Specialized networks have been developed and permit better follow-up of LS families, especially with respect to the endoscopy timelines. They encourage international, multicenter, prospective, observational studies, using the Prospective Lynch Syndrome Database (PLSD) [38], which allows the pooling of data from several collaborating European centers. “Real-life” data suggest that current management guidelines for Lynch syndrome should be more specific and be revised in light of the different gene- and gender-specific cancer risks.

### 3.2. Follow-Up of the Upper Digestive Tract

Regarding upper digestive tract management, despite a low level of evidence, most recommendations are in favor of the absence of routine surveillance for the small bowel and stomach. These explorations should take place only if the patient is symptomatic or if there is an upper digestive tract lesion in the family.

#### 3.2.1. Gastric Cancer

Individuals with LS have an overall lifetime cumulative risk of 0.7–13% of developing gastric cancer [39]. There is a trend toward higher prevalence of gastric cancer in carriers of *MLH1* and *MSH2* mutations compared with carriers of *MSH6* mutations [39]. Three observational studies provide information about upper gastrointestinal endoscopy monitoring in LS. In the first one, no gastric cancer was diagnosed, but only 32 gastroscopies were performed in 21 patients (32% of the population) during a ten-year follow-up [40]. In the second study, including 443 patients, gastric endoscopy was done in about 30% of cases; eight gastric cancers were identified and the rate of *Helicobacter pylori* (HP) infection did not differ from the general population [41]. The last work is a comparative study, where a single upper gastrointestinal endoscopy was proposed in both *MLH1* mutation carriers (*n* = 73) and mutation-negative family members (*n* = 32) [42]. The rates of HP infection, intestinal atrophy, and metaplasia were similar in both groups. No gastric neoplastic lesion was detected in either group, and only one case of duodenal cancer was detected in the mutation-positive group [42].

In conclusion, there is no convincing evidence of the usefulness of gastric surveillance in patients with LS. However, it appears mandatory to screen LS patients for HP, with subsequent eradication therapy if present. The HP infection rates seem effectively similar in LS patients and the general population [41,42]. The eradication of HP has reduced the incidence of gastric cancer by 35% in the general population [43,44]. A major study involving 1632 patients showed that, among persons with HP infection who had a family history of gastric cancer in first-degree relatives, HP eradication treatment also reduces the risk of gastric cancer [45].

Owing to a low gastric cancer risk in published series, routine surveillance does not seem necessary, but Helicobacter pylori screening should be systematically performed in patients with Lynch syndrome.

#### 3.2.2. Small Bowel Adenocarcinoma

In LS families, small bowel tumors are located mainly in the duodenum (43%) and the jejunum (33%). The cumulative lifetime risk of developing small bowel adenocarcinoma (SBA) has been estimated at 4.2% in patients with germline *MLH1* and *MSH2* pathogenic variants [46]. A recent study has reported gene-specific prospective cumulative cancer risks for duodenal adenocarcinoma in 3119 patients with LS. The risk of duodenal carcinoma was reported to be the highest for *MLH1* pathogenic variant carriers (6.5% for *MLH1* and 2.0% for *MSH2* carriers). No small bowel cancer was observed in patients with *MSH6* or *PMS2* mutations (462 and 124 patients, respectively, with mean follow-up of about 5 years) [47]. The median age of SBA diagnosis in LS patients ranged from 39 to 53 years [48,49,50]. Adenocarcinoma has been found in a large majority of cases (81% to 100%) [50,51]. It has also been suggested that LS-related SBA has a better prognosis than sporadic SBAs [47]. There is no demonstration of an excessive risk of SBA in the case of a first-degree relative with a history of SBA [52].

Two studies have evaluated video-capsule endoscopy (VCE) for small bowel neoplasia screening in LS, with prevalence of 8.6% of neoplasia (including adenomas and cancers) and 1.5% of cancers, respectively [53,54]. In most international guidelines, the frequency of cancer has appeared too low to justify systematic VCE screening. False-positive results have been found in at least 11% of patients, and are responsible for invasive secondary procedures such as balloon-enteroscopy or magnetic resonance enteroclysis [53].

The prospective study showed no SBA after average follow-up of 40 months in 35 patients with VCE screening, and compared the capsule to computed tomographic enteroclysis that missed two out of three cases of small bowel neoplasia [54]. Another retrospective study from the same team confirmed the limited benefit of VCE, detecting no small bowel neoplasia despite its repetition at an average interval of two years in 78% of the study population [55]. Finally, larger and prospective studies are required before drawing a definitive conclusion.

Interestingly, as most cancers are duodenal (with up to 6.5% cumulative risk in some mutation carriers), prospective studies evaluating gastroscopy with a longer endoscope (for examination of the proximal small bowel) would also be of interest.

There is currently not enough evidence to recommend routine small bowel monitoring, including by VCE. Larger studies are needed, and could also evaluate upper digestive endoscopy targeting the proximal and distal duodenum.

### 3.3. Follow-Up of the Lower Digestive Tract

#### 3.3.1. Endoscopic Aspect of Sessile Serrated Lesions and Colorectal Polyps in Lynch Syndrome

Usual colonic and small bowel adenomas are observed in LS, but it seems that sessile serrated polyps are also quite common in this disease. The role of sessile serrated lesions (SSL) and the serrated neoplasia pathway in LS is not fully understood. In one retrospective study, the frequency of SSL in LS patients was comparable to that of a matched general population group [56]. Non-polypoid adenomas (flat adenomas) are more frequently observed in LS patients than in people at average risk for CRC. SSLs are defined with a height of less than half the diameter, and advanced histology is defined by the presence of high-grade dysplasia or in situ carcinoma. In one study on 59 LS patients, adenomas were significantly more likely to be non-polypoid than they were in the 590 controls (43.3% vs. 16.9%, *p* < 0.001), and were particularly present in the proximal colon [57].

#### 3.3.2. Colorectal Cancer 


Age to start screening


The recommended follow-up begins at 20–25 years old for *MLH1* and *MSH2* pathogenic variant carriers and 30–35 years old for *MSH6* or *PMS2* mutation carriers, according to recent recommendations [17,28]. It is difficult to clarify, using evidence-based medicine, the appropriate age to start colonoscopic surveillance, even if this can be inferred from the individual risk of developing CRC in view of familial CRC history. However, several studies, even if not randomized, showed that this risk is dependent on the specific MMR gene mutated [58,59,60,61] *(*Table 3). A large international prospective cohort study involving over 3000 patients recently found cumulative incidence of CRC (at 75 years) of 46%, 35%, 49%, and 10% for *MLH1*, *MLH2*, *MSH6*, and *PMS2* mutation carriers, respectively, after a mean follow-up period of 7.8 years [47]. Most importantly, carriers of *MSH6* and *PMS2* mutations were at no or very low risk of CRC before the age of 40 years. Other studies have confirmed that the age of CRC onset in carriers of *MSH6* and *PMS2* mutations was delayed by 10 years compared with carriers of *MLH1* and *MLH2* mutations with negligible risk before the age of 40 [58,60]. The risk of developing advanced adenoma or CRC before 30 years old is extremely low in carriers of *MSH6* and *PMS2* mutations [60]. An annual colonoscopy in 155 males or 217 females in their 20s would prevent only one death by CRC [62]. Therefore, while it seems important to start colonoscopic surveillance at the age of 20–25 years for *MLH1* and *MLH2* mutation carriers, 30–35 years seems in some, but not all recent recommendations to be acceptable for *MSH6* and *PMS2* mutation carriers [15,16,17,28]. The limitation of these recommendation, once more, is the low level of evidence, especially based on prospective evaluation. That is why prospective series are still needed to precisely depict the risk of early/advanced cancer before 35 years in *MSH6* and *PMS2* patients.

The recommended age to start the follow-up is different according to the pathogenic variant carrier. It begins at 25 years old for *MLH1* and *MSH2* and 35 years old for *MSH6* or *PMS2* mutation carriers with a low CRC risk.


Colonoscopy intervals


An interval of 1 to 2 years between two high-quality surveillance colonoscopies is recommended in cases of LS. Without definitive scientific evidence, this question remains open and prospective studies are needed to determine whether some specific LS patients require a one-year interval. Different studies have analyzed intervals of 1, 2, or 3 years, which correspond to the practices of different European countries (1 year for Germany, 1–2 years for the Netherlands, and 2–3 years for Finland) [47,62,63,64,65,66].A large-scale international study involving more than 2700 LS patients with *MLH1*, *MLH2*, or *MSH6* mutations, out of a total of 16,000 colonoscopies, showed no difference in cancer incidence rates or CRC stage distribution according to the three surveillance modalities [67]. In retrospective cohort studies, the mean interval between colonoscopy and CRC diagnosis was between 24 and 36 months. This suggests the interest of more than one-year intervals [17,28,40]. On the other hand, the overall survival rates of patients diagnosed with interval CRC in surveillance programs are excellent, exceeding 90% [68,69,70].

In conclusion, precise stratification according to the type of MMR mutation is complex to implement, and a uniform interval of 2 years between each colonoscopy for any patient with LS is recommended. In high-risk CRC families, the colonoscopy interval could be one year, especially in *MLH1* and *MSH2* mutation carriers. Prospective studies are also underway and will be of importance.


Quality criteria for colonoscopy


There is no strong evidence of an increased risk of metachronous CRC in patients following polyp removal or CRC resection. The cumulative risk of metachronous CRC is highly variable according to the series, ranging from 2.1% in 7.8 years of follow-up to 16% in 10 years [31,61]. There is some evidence that incomplete removal of adenomas may significantly contribute to increased risk of CRC after colonoscopy [68]. However, recent data from the PLSD show that the impact of colonoscopy with polypectomy in preventing CRC is probably less than was previously thought [38]. Colonoscopic screening at 3-year intervals reduced the CRC rate by 62% on a controlled trial between LS patients with and without screening. The overall death rates were 10 versus 26 subjects in the study and control groups (*p* = 0.003) [37].

The effectiveness of a follow-up program depends on the quality of the colonoscopies performed: cleanliness, completeness, and possibly the use of chromoendoscopy [68,71,72]. Colonoscopy is evidently less effective for cancer prevention if the procedure does not reach the caecum or if bowel preparation is inadequate. In the case of colonoscopy with suboptimal bowel preparation (Boston bowel preparation scale < 6 or 8) or incomplete colonoscopy, colonoscopy must be repeated rapidly. Inadequate bowel preparation reduces the adenoma/advanced adenoma detection rate [73]. That said, the quality criteria for colonoscopies reported in recent studies are generally of a high standard and do not explain the incidence of CRC in LS cases during surveillance [74]. One possible explanation is that geography, environment, diet, and previous surgery, as well as age, gene, and gender, explain differences in the adenoma detection rate between the studies [38].


Chromoendoscopy versus high-definition white light endoscopy


Indigo-carmine chromoendoscopy (CE) as compared with white light endoscopy (WLE) was traditionally and is still recommended for the screening of LS patients. The optimal interval between colonoscopies should be based on the quality of the previous colonoscopy: optimal preparation, complete examination, and use of CE are associated with reduced CRC incidence [70]. The first studies comparing CE to WLE in LS clearly advantaged CE. Three monocenter studies and one multicenter study with back-to-back design and standard definition (SD) endoscopes demonstrated that CE was superior to WLE, reporting a WLE adenoma miss-rate ranging between 52 and 74% [75,76,77,78]. These studies are to be considered with caution as their methodology (back-to-back design) classically favors the second arm and can thus lead to overestimation of the CE effect on the WLE.

A recent randomized controlled trial showed slightly different evidence between CE and WLE. A study comparing a second examination with CE to a second with WLE showed no improvement in the detection of adenomas [79]. Another recent, randomized, parallel-group, multicenter study using high-definition endoscopes and experienced endoscopists showed a low and non-significant increase in adenoma detection by CE in more than 250 patients with LS (34.4% versus 28.1%, *p* = 0.28) [80]. Nevertheless, the detection rate of serrated lesions was higher with CE (37.5% versus 23.4%, *p* = 0.01). Yet, another study showed a higher detection rate of serrated lesions, but only in the proximal colon, using CE or WLE (adenoma detection rate of 33% versus 27%) [81]. The adenoma detection rate in the different studies are resumed in the Table 4.

Concerning virtual CE, there has been no clear demonstration of non-inferiority versus high-quality WLE. Some studies have shown superiority in Lynch syndrome [75,76], but others have shown inferiority versus dye-based CE [82,83,84]. High-quality colonoscopy has also been recognized as an essential component of successful cancer prevention in sporadic cases [85]. There is too little evidence to suggest that it may also be relevant for cancer prevention in LS patients. Quality performance indicators for colonoscopy in LS patients should at least meet or exceed those required for colonoscopy in sporadic cases. The caecal intubation rate and adenoma/polyp detection rate (>30% for experts from the ESGE) seem essential. The use of quality scores in colonoscopy reports should be recommended [86,87].

The challenge will be to develop the most efficient technology to improve the adenoma detection rate. The visual quality of endoscopes improves with each generation; the contribution of artificial intelligence is what will probably transform the methods of endoscopic monitoring in the years to come.

Colorectal dye-based chromoendoscopy is still recommended, but high-quality, high-definition white light endoscopy could be used according to some guidelines.

### 3.4. Pancreatic Cancer Risk

Even though the lifetime cumulative risk is below 5%, LS is one of the predisposing conditions to familial pancreatic cancer [88]. French recommendations [89] consider LS as a situation justifying surveillance in specific cases:If only one pancreatic cancer case in the family: surveillance only of first-degree relatives;If more than one pancreatic cancer case in the family: surveillance of all mutation carriers.

Screening for pancreatic cancer is recommended in these cases through annual pancreatic magnetic resonance imaging (MRI) and endoscopic ultrasonography in alternation, beginning at 45 years old or 10 years before the youngest case in the family. There is still limited evidence to support these guidelines and each context must be discussed with the patient [90,91].

We propose in the Table 5 a summary of the different follow-up guidelines in the Lynch syndrome.

## 4. Treatment of Colorectal Cancer in Lynch Syndrome

Large intestinal lesions are now treated by endoscopy by means of endoscopic submucosal dissection (ESD) [92,93,94]. Even though some clinical cases report early CRC treated by ESD [95], surgical treatment is clearly the recommended option in invasive CRC, especially for lymph node dissection. For LS patients with *MLH1* or *MSH2* mutations who develop CRC, even with adequate follow-up, the decision to perform segmental versus total/near-total colectomy should take into account the risks of metachronous CRC, the functional consequences of surgery, compliance with colonoscopy screening, and patient age and preferences. In most cases, segmental colectomy is recommended. For LS patients with *MSH6* or *PMS2* mutations, there is insufficient evidence to perform total/near-total colectomy rather than segmental resection. This is a strong recommendation by the last scientific society to evaluate the subject [17]. In addition, when abdominal perineal excision can be avoided, standard low anterior resection is a reasonable option to treat rectal cancers in LS patients, even though the residual colon is at risk of metachronous neoplasia [17].

## 5. Patient Adherence in Endoscopic Follow-Up Program

The LS patient’s adherence in repeated colonoscopies is challenging, but necessary to prevent colorectal neoplasia. While most individuals continue to engage in follow-up programs over the long term, about 20% have a partial or complete rupture with endoscopic follow-up [96]. In addition, a quarter of them require psychosocial support because of developing moderate depressive symptoms [97]. A clear and repeated explanation of the value of endoscopic surveillance in effectively preventing CRC risk is the key to successful adherence to surveillance programs. Specialized programs to remind patients of the dates of exams and follow-up are useful to avoid delays in colonoscopies.

The impact of a lifestyle change on people with familial risk is being evaluated and the concerned individuals may benefit from an explanation of these modifiable risk factors in order to adapt their lifestyle and, thereby, potentially reduce their level of adenoma/CRC risk [98,99,100]. Furthermore, there is evidence that smokers, particularly men with *MLH1* mutation and overweight/obesity, have an increased risk of CRC [101,102]. Nevertheless, the effect of patient education on increased adherence to the endoscopic follow-up program is not clearly proven. In a study on family communication in LS, patients who received educational resources had a higher likelihood of following up with a doctor and pursuing genetic testing than families without educational resources [103]. Educational workshop and support groups in LS have been evaluated in one study and a large majority of their participants perceived them as really helpful [104]. The spread of such programs could be one of the solutions to motivate LS patients to join the endoscopic surveillance program.

To improve patient adherence to endoscopic follow-up programs, it seems necessary to explain the benefits of the colonoscopies on their future life and to include them in dedicated education programs.

## 6. Conclusions

The first challenge is to properly identify patients with Lynch syndrome. Systematic MSI testing and/or loss of expression of MMR proteins in IHC for all CRCs is probably one of the most efficient approaches to be developed. Once the diagnosis has been clarified, several guidelines for screening and follow-up programs are now available. Discrepancies in these recommendations exist, with still limited prospective scientific evidence on some points. After reviewing the various literature and studies, we could nevertheless propose some clear recommendations for Lynch syndrome patients.


Routine gastric surveillance does not seem necessary owing to the low gastric cancer risk.HP screening should be systematically performed.There is currently not enough evidence to recommend routine small bowel monitoring, including by VCE.The recommended follow-up for colonoscopy begins at 20–25 years-old for *MLH1* and *MSH2* pathogenic variant carriers and 30–35 years-old for *MSH6* or *PMS2* mutation carriers.A uniform interval of 2 years between each colonoscopy is recommended. In high-risk CRC families, the colonoscopy interval could be one year, especially in *MLH1* and *MSH2* mutation carriers.Colorectal dye-based chromoendoscopy is still recommended, but high-quality, high-definition white light endoscopy could be used according to some guidelines.Surgical treatment is still clearly the recommended option in the case of invasive CRC.The patients should be followed in appropriate centers and specialized networks.Improved adherence of the patient to a screening program also seems essential.


The technology is constantly improving, as is the adenoma detection rate. The visual quality of endoscopes improves with each generation, and virtual dyes and artificial intelligence will probably transform our practices and endoscopic monitoring in the years to come.

## Figures and Tables

**Table 1 cancers-13-03505-t001:** Amsterdam II and Bethesda revised criteria.

Amsterdam II Criteria
Patients with the following four criteria:
At least three subjects with HNPCC narrow spectrum, one of whom is related to the other two in the first degree
At least one cancer diagnosed before the age of 50
At least two successive generations concerned
Exclusion of familial polyposis
**Revised Bethesda Criteria**
Patient with CRC diagnosed before age 50
Patient with CRC with microsatellite instability and/or loss of MMR protein expression in IHC before the age of 60 years
Patient with two synchronous or metachronous cancers belonging to the HNPCC broad spectrum regardless of age
Patient with a CRC and two or more first- or second-degree relatives with HNPCC broad-spectrum regardless of age
Patient with CRC and a first-degree relative with broad-spectrum HNPCC diagnosed before age 50

CRC: colorectal cancer; HNPCC: hereditary non-polyposis colorectal cancer; IHC: immunohistochemistry; MMR: mismatch repair system.

**Table 2 cancers-13-03505-t002:** Indications for immunohistochemistry (IHC) and/or microsatellite instability (MSI) testing.

American and British Guidelines [8,9,10]
Any patient with new colorectal cancer diagnosed
**European Guidelines** [18]
Any patient under 70 years of age presenting a first CRC

**Table 3 cancers-13-03505-t003:** Lifetime risks of CRC at 70 years old [4,47,59].

Type of Gene Mutation Carrier	Estimated Lifetime CRC Risk
*MLH1* and *MSH2*	Ranges from 40 to 52%
*MSH6*	Approximately 15%
*PMS2*	Between 3 and 13%

**Table 4 cancers-13-03505-t004:** Adenoma detection rate in the different studies with indigo-carmine chromoendoscopy or white light endoscopy.

Adenoma Detection Rate (%) or Total Number of Adenoma Detectedin Colonoscopy in the Different Studies	With Indigo-Carmine Chromoendoscopy	With White Light Endoscopy	*p*-Value
Reference [75], Perrod et al.	99/353 (28%)	60/211 (28.4%)	*p* > 0.05
Reference [76] Lecomte et al.	10/33 (30%) proximal colon only)	3/33 (9%) proximal colon only	*p* = 0.045
Reference [77], Hüneburg R et al.	13/47 (27%)	7/47 (14%)	no significant difference
Reference [78], Hurlstone et al.	52/16	24/13	*p* = 0.001
Reference [79], Rahmi et al.	32/78 (41%)	18/78 (23%)	*p* < 0.001
Reference [80], Stoffel et al.	5/28 (17%)	7/26 (26%)	no significant difference
Reference [81], Rivero-Sánchez et al.	34.4% on 128 patients	28.1% on 128 patients	no significant difference

**Table 5 cancers-13-03505-t005:** Summary of follow-up guidelines in Lynch syndrome cases.

Indication for Surveillance	Category	Modality	Age to Start (years)	Intervals	Use of Chromoendoscopy
**ESGE and British guidelines** [17,28]	*MLH1* and *MSH2* gene carriers	Colonoscopy	25	2 yearly until age 75 years old	ESGE guidelines: The use of chromoendoscopy may be of benefit in individuals with Lynch syndrome undergoing colonoscopy; however, routine use must be balanced against costs, training, and practical considerations.
	*MSH6* and *PMS2* gene, carriers	Colonoscopy	35	2 yearly until age 75 years old	
	Stomach, small bowel, and pancreas	No routine surveillance beside research protocols. Screening and eradication of Helicobacter pylori.	None	None	British guidelines: High-quality, high-definition white light endoscopy is the preferred modality for colonoscopy surveillance
**American guidelines** [15,16]	Colon, all mutation carriers	Colonoscopy	20 to 25, or 5 years before the youngest age of diagnosis of colorectal cancer in an affected family member	Every 1 to 2 years	No recent statement on the subject.
	Stomach, small bowel, and pancreas	No routine surveillance outside clinical trial. Screening and eradication of Helicobacter pylori.	None	None	

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
