# Peer review of "Endoscopy to Diagnose and Prevent Digestive Cancers in Lynch Syndrome"

_cancers, 2021, doi:10.3390/cancers13143505_

Round 1

Reviewer 1 Report

The manuscript „Endoscopy to diagnose and prevent digestive cancers in Lynch syndrome“ is a review on a hereditary colorectal cancer syndrom that associates with other tumors along and outside the GI-tract. It is defined by germline mutations in MMR-genes resulting in MSIhigh tumors.      The paper covers an interesting topic, but there are several shortcomings that should be addressed.  Overall, the manuscript seems rather hastily composed and better/more care is advised in particular with respect to clarity and structure. The are several guidelines on the topic worlwide and their recommendations vary partly substantially from the ESGE guidelines and the UK guidelines that the authors mainly refer to. This should be made clearer and better discussed. Overall, a more endoscopically centered (opinion) review could be expected based on the title of the paper.

As a reader of a review on that topic, I would appreciate a short comparison or a clear summary of the published guidelines or a paper higlighting additional aspects. Otherwise, I would rather refer to the guidelines themselves.

Some specific comments:

  • the chapters and the text only partly mirror the headline
  • some of the chapters are mostly a line up of studies, more grouping/commenting on relevance could be helpful
  • the logical structure of the manuscript should be improved, e.g. poor hierarchy in chapter 3.
  • referring to the title, the review could be better focus on the role of endoscopy in Lynch (diagnosis, screening, prevention, therapy...)
  • please add summaries/key statements at the end of each chapter
  • table 1 is difficult to read
  • table 2 should be improved: lower part does not correspond with upper part interms of structure
  • additional tables highlighting key issues are encouraged (tumor risk at different sites over time depending on the genetics; comparison of discrepant recommendations between guidelines)
  • introduction: the therapeutical significance of MSI in CRC and other tumor entities should mentioned (prognostic value, predictive value, checkpoint inhibtion, as risk information for relatives)
  • please update literature search -> important papers related to the topic are missing, e.g. Choi, I. J., et al. (2020). NEJM 382(5): 427-436.
  • chapter 4 is not at all on endoscopy
  • the positive interpretation of the CAPP2 trial is debatable, e.g. the primary endpoint was missed
  • chapter 5 needs revision with more careful statements
  • reading is made difficult by multiple typos -> please check

Author Response

We have cited the lines affected by the changes, the number of the lines correspond to the number without the "Track Changes" mode. 

Reviewer(s)' Comments to Author: 

Reviewer: 1

The manuscript “Endoscopy to diagnose and prevent digestive cancers in Lynch syndrome” is a review on a hereditary colorectal cancer syndrom that associates with other tumors along and outside the GI-tract. It is defined by germline mutations in MMR-genes resulting in MSI high tumors. The paper covers an interesting topic, but there are several shortcomings that should be addressed.  Overall, the manuscript seems rather hastily composed and better/more care is advised in particular with respect to clarity and structure. The are several guidelines on the topic worlwide and their recommendations vary partly substantially from the ESGE guidelines and the UK guidelines that the authors mainly refer to. This should be made clearer and better discussed. Overall, a more endoscopically centered (opinion) review could be expected based on the title of the paper.

As a reader of a review on that topic, I would appreciate a short comparison or a clear summary of the published guidelines or a paper highlighting additional aspects. Otherwise, I would rather refer to the guidelines themselves.

We thank the reviewer for this relevant comments. We have largely modified the manuscript to make it clearer for more clarity and structureand to take into account the comments of the reviewer. We have added other guidelines (American guidelines, line 112) and described and discussed the discordances between them. We have therefore summarized all the different guidelines in the table 5 add new tables (table 2 and 4) and drawn up a clear summary of the surveillance strategy that we believe should be adopted in different key statements at the end of each chapter, lines 145-147, 210-213, 256-260, 304-307, 338-343, 414-416, 509-510. The review is also now more focused on endoscopic management of Lynch syndrome patients.

Some specific comments:

the chapters and the text only partly mirror the headline

We have corrected some of the headlines and amended the text and chapters to better reflect them, in particular chapter 2.1, 3.3, 4.

some of the chapters are mostly a line up of studies, more grouping/commenting on relevance could be helpful

We have summarized the different studies and guidelines to make more relevant summary for the readers, in the tables and the key statements at the end of the chapter (lines 145-147, 210-213, 256-260, 304-307, 338-343, 414-416, 509-510). In addition, for more clarity we modified and add some tables (table 2: page 3 line 103, table 3: page 8 line 310, table 4: page 9 line 419 and table 5: page 11 line 444).

the logical structure of the manuscript should be improved, e.g. poor hierarchy in chapter 3.

We have modified the structure of the article, especially by better structuring chapter 3 and the sub-chapters (lines 167-174, 203-212, 256-260, 283-290, 300-307, 338-343, 409-416. In addition, hierarchy of chapter 3 and chapter 4 have also been modified.

referring to the title, the review could be better focus on the role of endoscopy in Lynch (diagnosis, screening, prevention, therapy...)

We removed some paragraphs (chapter 4-1) and focused this review on the role of endoscopy in Lynch syndrome throughout the manuscript. We have added Chapter 5 page 9, lines 483-507, on “Patient adherence in endoscopic follow-up program”.

please add summaries/key statements at the end of each chapter

We have added summaries/key statements at the end of each chapter to clarify the manuscript, lines 145-147, 210-213, 256-260, 304-307, 338-343, 414-416, 509-510.

table 1 is difficult to read

Table 1 has been modified to facilitate its understanding.

table 2 should be improved: lower part does not correspond with upper part interms of structure

Table 2 is now Table 5 “Summary of follow-up guidelines in Lynch Syndrome” which has been completely modified. We have changed it in accordance with other reviewer’s comments to add the guidelines and easily compared them.

additional tables highlighting key issues are encouraged (tumor risk at different sites over time depending on the genetics; comparison of discrepant recommendations between guidelines)

We added three tables for highlighting discrepancies according the different guidelines. Table 2, explains indications for MMR immunohistochemistry (IHC) or microsatellite instability (MSI) tests. Table 3: the lifetime risks of CRC at 70 year-olds. Table 4 clarifies the guidelines for chromoendoscopy use. Table 5 summarize the endoscopic follow-up guidelines in Lynch syndromeincluding modality, age to start and interval of screening and according the MMR gene mutated. (table 2: page 3 line 103, table 3: page 8 line 310, table 4: page 9 line 419 and table 5: page 11 line 444).

introduction: the therapeutical significance of MSI in CRC and other tumor entities should mentioned (prognostic value, predictive value, checkpoint inhibtion, as risk information for relatives)

We thank the author for this relevant comment, we have added some sentences to summarize the therapeutical significance of MSI in CRC (line 40 to 54).

please update literature search -> important papers related to the topic are missing, e.g. Choi, I. J., et al. (2020). NEJM 382(5): 427-436.

We have updated literature search and added some reference throughout the manuscript (ref 45, line206) and we have changed the sentences accordingly.

chapter 4 is not at all on endoscopy

We are agreed with this comment and we decided to remove chapter 4concerning prevention of digestive cancers in Lynch syndrome. In the paragraph 4 we now only highlighted endoscopic and surgical treatment of Lynch related colorectal cancer. We have added Chapter 5 page 9, lines 483-507, on “Patient adherence in endoscopic follow-up program”.

the positive interpretation of the CAPP2 trial is debatable, e.g. the primary endpoint was missed

This point was now deleted since it is not at all on endoscopy.

chapter 5 needs revision with more careful statements

We have rewritten this chapter (now it is chapter 6) to include more careful statements in line of this review focused on endoscopy, line 513 to 522.

reading is made difficult by multiple typos -> please check

We apologize for the many typos and misspellings and the manuscript has been carefully review by Jeffrey Arsham a native English speaker.

Reviewer 2 Report

This is an interesting document and a useful summary of the evidence of clinical management strategies for GI cancers associated with Lynch syndrome.

Overall however I feel it needs to clarify some of the statements made within, and importantly, provide a strategy for how paradigms in surveillance might be changed and developed, rather than simply discussing existing methodology, with greater discrimination and discussion of short-medium term possibilities.

There are many typos and misspellings throughout the document. e.g. "briad", I have not listed them all

Abstract/Summary: Needs rewriting, it is not clear. Typo: “Few g Guidelines from different….” “we will resume the criteria to suspect” meaning unclear

Criteria to suspect section:

The introducing paragraph is not clear

Clinical suspicion section

What are the relative benefits or risks of each approach?  A review should be more discriminating perhaps?

"MSI testing for all CRCs is becoming a gold standard:"  I disagree - Why MSI and not IHC to test for MMR status? Needs explaining, given competing evidence suggests IHC may be at least equivalent if not superior

Somatic analyses section

Where is the discussion of somatic testing post constitutive testing?  ie. for LLS

"It is recommended to use omly one test if there is a low suspicion of LS (e.g., a single case of LS cancer spectrum be-fore age of 60), but to use both tests if there is a high suspicion of LS (e.g., multiple LS cancer spectrum in the family)"

Why is this?  What is the rationale?

"It is worth noting that these tests should performed on cancers and not adenomas since false negative results exist."

On what basis is the statement made? dMMR is an early event in carcinogenesis so should be visible in even low grade adenomas.

Endoscopic screening section

"One major point is that patients with LS should be followed in expert centers with specialized networks." I agree with this statement but in the document the specific value of networks is not discussed.

Endoscopic aspect section:

Needs discussion in context of PLSD data, more discussion about quality factors in colonoscopy

Author Response

Point by point response to the reviewers

We have cited the lines affected by the changes, the number of the lines correspond to the number without the "Track Changes" mode.

Reviewer(s)' Comments to Author:

Reviewer: 2

His is an interesting document and a useful summary of the evidence of clinical management strategies for GI cancers associated with Lynch syndrome.

Overall however I feel it needs to clarify some of the statements made within, and importantly, provide a strategy for how paradigms in surveillance might be changed and developed, rather than simply discussing existing methodology, with greater discrimination and discussion of short-medium term possibilities.

We thank the reviewer for these relevant comments concerning the clarity, structure and hierarchy of the manuscript also highlighted by other reviewers. We have therefore better summarized all the recommendations from the different guidelines in the text, in key statement et the end of each chapter (lines 145-147, 210-213, 256-260, 304-307, 338-343, 414-416, 509-510) and by adding some tables (Table 2: page 3 line 103, table 3: page 8 line 310, table 4: page 9 line 419 and table 5: page 11 line 444).

We have drawn up a clear summary of the follow-up strategy of Lynch syndrome patients that we believe should be adopted (lines 145-147, 210-213, 256-260, 304-307, 338-343, 414-416, 509-510). We also discussed the short-medium term possibilities of Lynch syndrome management, especially with new endoscopic techniques (line 409-413).

here are many typos and misspellings throughout the document. e.g. "briad", I have not listed them all

We apologize for the many typos and misspellings; a careful review has been carried out by an English native speaker.

Abstract/Summary: Needs rewriting, it is not clear. Typo: “Few g Guidelines from different….” “we will resume the criteria to suspect” meaning unclear

We have removed the typos in all the manuscript. We have also rewritten the abstract and the summary to make them clearer.

Criteria to suspect section:

The introducing paragraph is not clear

We have reworded the Introduction and added some points as suggested by reviewer 1.

Clinical suspicion section

What are the relative benefits or risks of each approach?  A review should be more discriminating perhaps?

Thank you for this relevant comment. Another review in this special issue is focusing on this topic (Diagnosis of Lynch syndrome and strategies to distinguish Lynch-related tumors from sporadic MSI/dMMR tumors” Leclerc J. et al, reference 21), so we have chosen not to explore it further. Nevertheless, we have modified the paragraph “Clinical suspicion of Lynch Syndrome” to make it clearer and specify the criteria for suspecting Lynch syndrome.

"MSI testing for all CRCs is becoming a gold standard:"I disagree - Why MSI and not IHC to test for MMR status? Needs explaining, given competing evidence suggests IHC may be at least equivalent if not superior

We misspoke on this point. We would like to say that an universal testing for all CRCs is becoming a gold standard either by MSI molecular test and/or expression of MMR proteins by IHC. This point has been corrected (line 91 to 94).

Somatic analyses section

Where is the discussion of somatic testing post constitutive testing?  ie. for LLS

We effectively did not address the situation of somatic testing after negative constitutive MMR analysis as this review focus on endoscopy and digestive cancers in Lynch syndrome. This situation a rare and out of the focus of this review in our opinion. Nevertheless, we add a sentence concerning Lynch like syndrome in this paragraph (line 119 to 126).

"It is recommended to use only one test if there is a low suspicion of LS (e.g., a single case of LS cancer spectrum be-fore age of 60), but to use both tests if there is a high suspicion of LS (e.g., multiple LS cancer spectrum in the family)"Why is this?  What is the rationale?

We have reworded this paragraph. Most guidelines recommend to perform only one test (MSI test or MMR IHC test) for a universal testing of CRC. Nevertheless, in the case of LS suspicion both tests are recommended in order not to miss a LS since discordances between the two tests can exist. Both tests for all CRCs seems difficult to carry out routinely because of their cost and they are time consuming. These points are now better explained in the manuscript (line 127 to 143).

"It is worth noting that these tests should performed on cancers and not adenomas since false negative results exist."

On what basis is the statement made? dMMR is an early event in carcinogenesis so should be visible in even low grade adenomas.

You are absolutely right, 60% of adenomas larger than one centimeter have an dMMR/MSI status in the LS. Nevertheless, some adenomas have a false negative result and whenever possible, testing should be performed on CRC and not on adenoma. This point is now clarified in the manuscript (line 141 to 143).

Endoscopic screening section

"One major point is that patients with LS should be followed in expert centers with specialized networks." I agree with this statement but in the document the specific value of networks is not discussed.

We are totally agreeing, we have therefore discussed this point, in particular by discussing about the PLSD and its contribution (line 167 to 174 and 351 to 358).

Endoscopic aspect section:

Needs discussion in context of PLSD data, more discussion about quality factors in colonoscopy

We have now included some important observations from the PLSD concerning endoscopic screening and completed the discussion on the quality factors in colonoscopy (line 348 to 371)).

Reviewer 3 Report

Olivier and colleagues present a review on the current status of guidelines to detect and surveil individuals with Lynch syndrome. Though an interesting topic and evolving area, this manuscript requires significant and substantial editing.

Overall comments

  • Please carefully review for grammatical and textual errors. There are too many to list in the comments.
  • From my reading, many words are incorrectly used in context. Please thoroughly read the manuscript to ensure that appropriate words/terms are used in context
  • The review needs to be reorganized for better flow. As it stands, the subsection organization doesn’t make much sense in terms of the content, which makes it hard to follow the argument the authors are trying to make

Summary/Abstract

  • Remove ‘g’ before Guidelines
  • Summary: There are quite a few guidelines available for the surveillance of individuals with Lynch syndrome (e.g. American Gastroenterological Association, CDC, American Society of Colon and Rectal Surgeons), so the initial argument that there are not available guidelines is inherently flawed. The authors need to take a different approach
  • Abstract: ‘Resume’ is not the correct word to use here

Introduction

  • LS has a population prevalence is roughly of 1 in 280. Please add references for your statistics.
  • Italicize genes, e.g. MLH1
  • There is no transition between the description of the MMR system and the purpose of the review. This needs to be rewritten to lead into the goal of the review. Understanding MMR isn’t really necessary for this review and can be edited down

2. Criteria to suspect

  • Edit for grammar, clarity, readability

2.1

  • IHC is not recommended for all CRCs, only for those diagnosed with CRC that meet specific criteria (see Revised Bethesda guidelines). Please revise for accuracy.
  • Please add information on how clinicians use the LS criteria. This will better inform your argument, rather than just stating that guidelines exist.

2.2

  • Please spell out dMMR (MMR deficiency) prior to use of the acronym
  • Include a reference for discordant methodology
  • Last sentence – include a reference

3.

  • Section is very difficult to read and requires extensive editing

4.

  • I don’t think there’s a need to discuss risk factors for sporadic CRC, as that is not the focus of this review. This content can be removed to focus on LS-associated CRC. If this content is desired as a comparison/contrast, it could be incorporated into a figure rather than in the text.
  • Similar to Section 3, this section is difficult to read and requires extensive editing.
  • A recent study by Vilar-Sanchez et al in 2020 adds to the literature surrounds LS and NSAIDs. This can be included in this section
  • Please remove or define BSG. It seems out of context

Conclusion

  • There is no discussion of patient education (discussed in the Conclusion). The authors should include this into the review to make it more comprehensive.
  • ‘Complete adherence of the patient is therefore essential’. I agree, adherence is very important, but suggesting complete adherence is unrealistic. As this is a review, the authors should suggest/discuss ways to improve adherence and the feasibility to these approaches. Adding this content will improve the importance/urgency of the review.

Author Response

Point by point response to the reviewers

We have cited the lines affected by the changes, the number of the lines correspond to the number without the "Track Changes" mode.

Reviewer(s)' Comments to Author:

Reviewer: 3

Olivier and colleagues present a review on the current status of guidelines to detect and surveil individuals with Lynch syndrome. Though an interesting topic and evolving area, this manuscript requires significant and substantial editing.

Overall comments

Please carefully review for grammatical and textual errors. There are too many to list in the comments.

From my reading, many words are incorrectly used in context. Please thoroughly read the manuscript to ensure that appropriate words/terms are used in context.

We apologize for the many typo errors and misspellings; a careful review has also been carried out by an English native speaker.

The review needs to be reorganized for better flow. As it stands, the subsection organization doesn’t make much sense in terms of the content, which makes it hard to follow the argument the authors are trying to make

According to your comment and those of the others reviewers we have corrected some of the headlines and modified the organization of this review to obtain of a better flow for the readers. In addition, we have amended the text and chapters to better reflect the titles of the subsection, in particular chapter 2.1, 3.3 and 4. We have modified the structure of the manuscript, especially by better structuring chapter 3 and the sub-chapters. In addition, hierarchy of chapter 3 and chapter 4 have also been modified.

We have added key statements at the end of the chapter (lines 145-147, 210-213, 256-260, 304-307, 338-343, 414-416, 509-510). In addition, for more clarity we modified and add some tables (table 2: page 3 line 103, table 3: page 8 line 310, table 4: page 9 line 419 and table 5: page 11 line 444).

Summary/Abstract

Remove ‘g’ before Guidelines

This typo error has been removed.

Summary: There are quite a few guidelines available for the surveillance of individuals with Lynch syndrome (e.g. American Gastroenterological Association, CDC, American Society of Colon and Rectal Surgeons), so the initial argument that there are not available guidelines is inherently flawed. The authors need to take a different approach

We thank the reviewer for this relevant comments and the summary has been modified. In addition, as suggested by the other reviewers we have added some other guidelines (American Gastroenterological Association, National Comprehensive Cancer Network guidelines) in the manuscript. We also take a different approach bydescribing and discussing the discordances between the different guidelines to propose an up a clear summary of the recommendations and the follow-up strategy that we believe should be adopted.

Abstract: ‘Resume’ is not the correct word to use here

We have changed the term “resume” for “recall”; The abstract is now rewriting according to the different reviewer’s comments.

Introduction

LS has a population prevalence is roughly of 1 in 280. Please add references for your statistics.

We have changed the prevalence and have added two references (line 32, references 3 and 4.).

Italicize genes, e.g. MLH1

We have italicized all the genes in the manuscript.

There is no transition between the description of the MMR system and the purpose of the review. This needs to be rewritten to lead into the goal of the review. Understanding MMR isn’t really necessary for this review and can be edited down

We have shortened the description of the MMR system and we focused the Introduction on the purpose of the review, provide a clear summary of the guidelines and the follow-up strategy of Lynch syndrome-associated digestive cancers. We have also highlighted the therapeutic impact of dMMR/MSI status in CRC as suggested by the reviewer 1 (line 40 to 54).

  1. Criteria to suspect

Edit for grammar, clarity, readability

We have rewritten all this paragraph which is now included in the chapter 2.1. Starting line 67.

2.1

IHC is not recommended for all CRCs, only for those diagnosed with CRC that meet specific criteria (see Revised Bethesda guidelines). Please revise for accuracy.

The recommendations for MMR, IHC/MSI tests in CRC vary according the guidelines but the most recent guidelines tend to a universal testing of all CRC.We have revised this chapter for accuracy (line 85 to 94 and 127 to 147).

Please add information on how clinicians use the LS criteria. This will better inform your argument, rather than just stating that guidelines exist.

Thank you for this relevant comment. We have added that at least all CRC that meet the revised Bethesda criteria must have a IHC MMR/MSI tests. Due to LS screening and the high impact of dMMR/MSI status, like recent some guidelines, we proposed a universal screening of all CRC (line 127 to 147).

2.2

Please spell out dMMR (MMR deficiency) prior to use of the acronym

The acronym has been defined (line xx).

Include a reference for discordant methodology

We have included now three references (references 22, 23, 24, line 129).

Last sentence – include a reference

This sentence has been modified according to the comment of the reviewer 2 and a reference has been added (line 137 to 139, reference 27).

3.

Section is very difficult to read and requires extensive editing

We have made many changes in this section, including better structuring chapter 3 and the sub-chapters as also suggested by reviewer 1 and 2.

4.

I don’t think there’s a need to discuss risk factors for sporadic CRC, as that is not the focus of this review. This content can be removed to focus on LS-associated CRC. If this content is desired as a comparison/contrast, it could be incorporated into a figure rather than in the text.

Similar to Section 3, this section is difficult to read and requires extensive editing.

A recent study by Vilar-Sanchez et al in 2020 adds to the literature surrounds LS and NSAIDs. This can be included in this section.

We agree and we have removed the section 4.1 because it is not on the focus of this review, including the paragraph on NSAIDs as suggested by reviewer 1. As also suggested by two other reviewers we have performed many changes in the section 4, to make it clearer for the readers and focused on the treatment of colorectal neoplasia in LS.

Please remove or define BSG. It seems out of context

We have removed it.

Conclusion

There is no discussion of patient education (discussed in the Conclusion). The authors should include this into the review to make it more comprehensive.

‘Complete adherence of the patient is therefore essential’. I agree, adherence is very important, but suggesting complete adherence is unrealistic. As this is a review, the authors should suggest/discuss ways to improve adherence and the feasibility to these approaches. Adding this content will improve the importance/urgency of the review.

We have added a full chapter to discuss ways to improve adherence of the patient (chapter 5, line 484 o 510).

Round 2

Reviewer 1 Report

We would like to thank the authors for their revisions and corrections which have improved manuscript.

I recommend to revise the following point:

line 45/46: „…in the event of MMR mutation their relatives must 46 likewise be tested.“ This statement is not in line with human ethics.  Genetic testings cannot and shouldn´t be made for mandatory for any individual.

Author Response

Point by point response to the reviewers

Reviewer(s)' Comments to Author:

Reviewer: 1

We would like to thank the authors for their revisions and corrections which have improved manuscript.

We thank the reviewer for this comment, and for his valuable contribution in improving the manuscript.

I recommend to revise the following point:

line 45/46: „…in the event of MMR mutation their relatives must 46 likewise be tested.“ This statement is not in line with human ethics.  Genetic testings cannot and shouldn´t be made for mandatory for any individual.

You are perfectly right : we have changed the sentence by “their relatives should be offered the opportunity to be tested for the mutation”. Line 46-47.

Reviewer 2 Report

Thanks to the authors for their corrections and re-organisation of the papers.  This has made the paper much clearer and the context outlined in a helpful way.  

There are still a few typos which may need to be corrected e.g. 'estimating population frequency' should be 'estimated population frequency '

I am not sure what is meant by 'therapeutic attitudes'

Author Response

Point by point response to the reviewers

Reviewer(s)' Comments to Author:

Reviewer: 2

Thanks to the authors for their corrections and re-organisation of the papers.  This has made the paper much clearer and the context outlined in a helpful way.  

We thank the reviewer for this comment, and for his valuable contribution in improving the manuscript.

There are still a few typos which may need to be corrected e.g. 'estimating population frequency' should be 'estimated population frequency '

 We apologize again for the many typos and misspellings and the manuscript has been carefully review by ourselves and by Jeffrey Arsham a native English speaker. We have changed for “estimated population frequency” line 33.

I am not sure what is meant by 'therapeutic attitudes

We have clarified and have changed the sentence for “to propose the most consensual overall

Reviewer 3 Report

Similar to the original version, the manuscript needs to be closely proofread for grammatical errors

The authors try to include an incredible amount of information, which I think is to their detriment. Because of this, the primary message of each section/subsection is very hard to parse out. Rather than including information from each individual study, the authors need to clearly summarize and make a clear point.

Summary

  • Restructure and combine sentences on lines 12-14
  • Remove ‘a’ in line 15

Abstract

  • Line 18 – ‘Recall’ isn’t the correct word here. A better choice would be ‘summarize’
  • Line 22 – The discrepancies between guidelines already exist, therefore ‘may’ can be removed
  • Lines 23-24 – Conclusion sentence needs to be rewritten for clarity
  1. Introduction
  • Spelling out DNA isn’t necessary; it’s a common abbreviation
  • EPCAM needs to be spelled correctly and italicized
  • Rather than ‘known as MSI’, should be ‘resulting in MSI’
  • Lines 43-46 – sentence is poorly written and difficult to understand
  • Please be consistent in using abbreviations (e.g. MMR deficiency)
  • Line 46 – I don’t think it’s accurate to say ‘confer prognosis’, rather is associated with a better prognosis
  • dMMR and MSI aren’t necessarily interchangeable; please be consistent
  • The introduction should be shortened and focused. If the purpose of the review is LS, there needs to be a focus on LS. Describing the associations with dMMR and MSI is important, however it is necessary to tie that into why this information is important for LS.
  • Line 60-62 – Fragmented sentence
  1. Criteria to suspect and diagnose LS
  • Line 71 – I’m confused by the statement ‘they belong to the restricted spectrum’. Please clarify
  • Line 73 – ‘suspected’ instead of ‘suspect’
  • Line 75 – Explain why there is a focus on digestive cancers
  • Lines 77-84 – Why are the criteria most useful, or too restrictive? These are vague statements that need explanation
  • Line 82, 87 – spell out IHC at first use on line 82, not line 87
  • Line 92 – Table 2 should be in parentheses
  • Line 110 – Where is Table 2?
  • Line 120 – ‘Extinction’ is not an appropriate term here
  • ‘Oncogenetic consultation’ should be genetic counseling
  • Lines 134-150 – This paragraph needs to be much more concise. There are run-on sentences and redundancies.
  1. Endoscopic follow up
  • Section 3.1 is confusing and needs better organization to allow for the reader to understand the message of the section
  • Line 185 – Remove ‘in’
  • Line 194-195 – Remove sentence ‘There is no…”. It doesn’t add anything.
  • Lines 208-210 – Need reference
  • Choose between using the abbreviation HP or pylori; please be consistent
  • How does pylori prevalence vary between individuals with LS and the general population? Based on what is provided in the subsection, there really isn’t a difference, and therefore suggesting that H. pylori screening for LS doesn’t make sense
  • Lines 222-223, lines 232-234 – Redundant sentences, please revise
  • Lines 240-250 – This subsection needs to be significantly cleaned up
  • Rather than described each individual study of LS and carcinoma, it would be much cleaner and clearer if the authors provided a summary table of the different studies (gastric, small bowel, etc.) and then focus the text on the important findings from the most relevant studies. Currently, there is too much information to be able to discern the primary message of the authors
  • 3.a. – Put ‘sessile’ in the subsection title
  • Lines 312-314 – This is not a conclusion sentence, this is prime information that should be integrated into the subsection. The conclusion sentence should provide a summary of the subsection.
  • Table 3 – Provide column titles for the table
  • Lines 325-343 – Please clean up this subsection
  • Line 394 – Is this referring to RCTs?
  • Line 404-405 – Rather than ‘any superiority’, it should be non-inferior
  • Lines 382-424 – For non-clinicians, this subsection could be very confusing. Again, it would be beneficial to summarize information in a table, rather than trying to address everything in the text
  • Table 4 – I don’t think that the table adds much to the review. It could be helpful to clearly summarize all screening guidelines in a table, and then focus on the most relevant information for the text. This information can be integrated in Table 5
  1. Treatment of CRC in LS
  • Lines 479, 484 – Italicize genes
  1. Patient adherence
  • Line 501-502 – Remove ‘ has begun to be’ and replace with ‘is being evaluated’
  • Lines 517-518 – This sentence is unclear
  1. Conclusion
  • The conclusion section would be an opportune time to provide recommendations or suggestions to improve testing/detection/treatment, rather than just summarize the text. Adding this information would add novelty to the review.

Author Response

Point by point response to the reviewers

Reviewer(s)' Comments to Author:

Reviewer: 3

Similar to the original version, the manuscript needs to be closely proofread for grammatical errors

 We apologize again for the many typos and misspellings and the manuscript has been carefully review by ourselves and by Jeffrey Arsham a native English speaker. We have changed for “estimated population frequency” line 33.

The authors try to include an incredible amount of information, which I think is to their detriment. Because of this, the primary message of each section/subsection is very hard to parse out. Rather than including information from each individual study, the authors need to clearly summarize and make a clear point.

We thank the reviewer for this relevant comment. We have so tried to decrease the number of information by deleting many sentences, lines 38-39, 45-48, 134-137-138,141-144, 236-238, 244-245, 249-252, 258. According to your other comment (here below) we have create a new table to summarize information of the section 3 and delete the old table 4, we integrate it datas in the table 5. Finally we have clearly summarized and maked a clear point in each sub section and in the conclusion of the manuscript.

 Summary

Restructure and combine sentences on lines 12-14

Remove ‘a’ in line 15

We have restructure the sentences lines 12-14 and remove the “a”.

Abstract

Line 18 – ‘Recall’ isn’t the correct word here. A better choice would be ‘summarize’

Line 22 – The discrepancies between guidelines already exist, therefore ‘may’ can be removed

Lines 23-24 – Conclusion sentence needs to be rewritten for clarity

We have changed the sentences according to your comment lines 24-25.

  1. Introduction

Spelling out DNA isn’t necessary; it’s a common abbreviation

EPCAM needs to be spelled correctly and italicized

Rather than ‘known as MSI’, should be ‘resulting in MSI’

We have modified the sentence line 41.

Lines 43-46 – sentence is poorly written and difficult to understand

We have rewrite the sentence line 43-45.

Please be consistent in using abbreviations (e.g. MMR deficiency)

We have needed to use again “MMR deficiency” after write the abbreviation, cause it is the start of one sentence, line 49, we then have tried to be consistent and used the abbreviation.

Line 46 – I don’t think it’s accurate to say ‘confer prognosis’, rather is associated with a better prognosis. We have provided the change line 50.

dMMR and MSI aren’t necessarily interchangeable; please be consistent

We have change this fact for the line 57-58, for the gastric and small bowel adenocarcinoma, status, but for the others cases, dMMR and MSI both are possible in our opinion.

The introduction should be shortened and focused. If the purpose of the review is LS, there needs to be a focus on LS. Describing the associations with dMMR and MSI is important, however it is necessary to tie that into why this information is important for LS.

We have deleted some sentences to stay focus on the Lynch syndrome and to shorten the introduction. We delete a sentence lines 38-39 and lines 45-48.

Line 60-62 – Fragmented sentence

We have fragmented the sentence.

  1. Criteria to suspect and diagnose LS

Line 71 – I’m confused by the statement ‘they belong to the restricted spectrum’. Please clarify

We have tried to clarify line 77-78.

Line 73 – ‘suspected’ instead of ‘suspect’.

We have corrected.

Line 75 – Explain why there is a focus on digestive cancers

We have explain line 80.

Lines 77-84 – Why are the criteria most useful, or too restrictive? These are vague statements that need explanation

We have precised why the criteria are useful and restrictive lines 82-83 and 85-86.

Line 82, 87 – spell out IHC at first use on line 82, not line 87. Corrected.

Line 92 – Table 2 should be in parentheses. Corrected.

Line 110 – Where is Table 2?

There was an issue we have replaced the table 2  line 109.

Line 120 – ‘Extinction’ is not an appropriate term here. Replaced by “mutation”

‘Oncogenetic consultation’ should be genetic counseling. Corrected.

Lines 134-150 – This paragraph needs to be much more concise. There are run-on sentences and redundancies.

We have modified this paragraph by deleting some sentences (lines 134-137-138-141-144) and -rewording to make it more concise.

  1. Endoscopic follow up

Section 3.1 is confusing and needs better organization to allow for the reader to understand the message of the section

We reorganize the section and try to clarify the message, lines  169-171 and 180-182.

Line 185 – Remove ‘in’. It is done.

Line 194-195 – Remove sentence ‘There is no…”. It doesn’t add anything. It is done also.

Lines 208-210 – Need reference. We have added the reference 42 which corresponds line 209.

Choose between using the abbreviation HP or pylori; please be consistent. We used the abbreviation HP lines

How does pylori prevalence vary between individuals with LS and the general population? Based on what is provided in the subsection, there really isn’t a difference, and therefore suggesting that H. pylori screening for LS doesn’t make sense

We have changed this subsection lines 212-219 : according to the different studies that we have reported (references 41-42-45) the HP infection rates seems similar between LS patient and general population and his eradication seems efficient on reducing the gastric cancer risk in LS (reference 45) .

Lines 222-223, lines 232-234 – Redundant sentences, please revise

We have revised the second sentence to decrease the redundancy lines 236-238.

Lines 240-250 – This subsection needs to be significantly cleaned up

Rather than described each individual study of LS and carcinoma, it would be much cleaner and clearer if the authors provided a summary table of the different studies (gastric, small bowel, etc.) and then focus the text on the important findings from the most relevant studies. Currently, there is too much information to be able to discern the primary message of the authors

We have provided majors changes and delete many sentences to deliver less information and stay focus on a clear message, delete lines 236-238, 244-245, 249-252, 258.

3.a. – Put ‘sessile’ in the subsection title. We have provide the change line 274.

Lines 312-314 – This is not a conclusion sentence, this is prime information that should be integrated into the subsection. The conclusion sentence should provide a summary of the subsection.

We have changed the sentence to provide a better summary of the subsection lines 318-321. This prime information is available at the beginning of the subsection lines 291-293.

Table 3 – Provide column titles for the table. We apologize for this error and provide column title line 328.

Lines 325-343 – Please clean up this subsection. We have cleaned up and delete some sentences with minor importance lines 338-340 and 349-351.

Line 394 – Is this referring to RCTs? Yes, we have precised line 404.

Line 404-405 – Rather than ‘any superiority’, it should be non-inferior. We have provided the change line 414.

Lines 382-424 – For non-clinicians, this subsection could be very confusing. Again, it would be beneficial to summarize information in a table, rather than trying to address everything in the text

We have so created a new table (table 4)  to precise the information and the adenoma detection rate according to the different studies. Line 437

Table 4 – I don’t think that the table adds much to the review. It could be helpful to clearly summarize all screening guidelines in a table, and then focus on the most relevant information for the text. This information can be integrated in Table 5

We have deleted the “old” table 4 and add his information in the table 5 line 462.

  1. Treatment of CRC in LS

Lines 479, 484 – Italicize genes. It is done lines 473 and 478.

  1. Patient adherence

Line 501-502 – Remove ‘ has begun to be’ and replace with ‘is being evaluated’. We have proceeded to the change line 491.

Lines 517-518 – This sentence is unclear. We have reworded the sentence, lines 506-509.

  1. Conclusion

The conclusion section would be an opportune time to provide recommendations or suggestions to improve testing/detection/treatment, rather than just summarize the text. Adding this information would add novelty to the review.

We have completed the conclusion with all our recommendations to make a clear point, and finish by our suggestions concerning the future of the endoscopic follow up. Lines 518 - 544

Round 3

Reviewer 3 Report

With additional editing for grammar, sentence structure, and clarity, the authors have adequately addressed my comments.